# Discrete-Time Memristor Model for Enhancing Chaotic Complexity and Application in Secure Communication

**DOI:** 10.3390/e24070864

**Published:** 2022-06-23

**Authors:** Wenhao Yan, Wenjie Dong, Peng Wang, Ya Wang, Yanan Xing, Qun Ding

**Affiliations:** 1Electronic Engineering College, Heilongjiang University, Harbin 150080, China; yanwh0512@163.com (W.Y.); 2201740@s.hlju.edu.cn (P.W.); wangya1648@163.com (Y.W.); xingyanan063922@126.com (Y.X.); 2Beijing Aerospace Institute of Automatic Control, Beijing 100854, China; 1202873@s.hlju.edu.cn

**Keywords:** *TiO_2_*, memristor model, discrete time memristor, fixed point, secure communication

## Abstract

The physical implementation of the continuous-time memristor makes it widely used in chaotic circuits, whereas the discrete-time memristor has not received much attention. In this paper, the backward-Euler method is used to discretize the $TiO_{2}$ memristor model, and the discretized model also meets the three fingerprints characteristics of the generalized memristor. The short period phenomenon and uneven output distribution of one-dimensional chaotic systems affect their applications in some fields, so it is necessary to improve the dynamic characteristics of one-dimensional chaotic systems. In this paper, a two-dimensional discrete-time memristor model is obtained by linear coupling of the proposed $TiO_{2}$ memristor model and one-dimensional chaotic systems. Since the two-dimensional model has infinite fixed points, the stability of these fixed points depends on the coupling parameters and the initial state of the discrete $TiO_{2}$ memristor model. Furthermore, the dynamic characteristics of one-dimensional chaotic systems can be enhanced by the proposed method. Finally, we apply the generated chaotic sequence to secure communication.

## 1. Introduction

In 1971, Chua et al. theoretically predicted the existence of a nonlinear passive electronic component between charge and magnetic flux according to the theory of electronics, which they named the memristor, and proposed the concept of the generalized memristor [1,2]. For a long time, due to the lack of suitable materials, the research of memristors has been in the theoretical stage. Until 2008, HP Laboratory in the United States manufactured a solid memristor by using double-layer $TiO_{2}$ thin film, which made the memristor model physically realized [3]. In 2013, Adhikari and Biolek [4,5] proposed three fingerprints of the memristor, showing that a memristor can exhibit a pinched hysteresis loop when applying a bipolar sinusoidal current. When the frequency exceeds the critical value, the area of hysteresis sidelobe decreases monotonically with the increase of the frequency. As the frequency increases to infinity, the tight hysteresis loop of the device degenerates into a single-valued function. As the memristor is a nonlinear component and has memory capability, memristors are widely used in neural networks [6,7,8], nonlinear circuit systems [9,10], nanotechnology [11], electrical engineering, [12,13] and other fields.

Chaos is an inherent randomness of deterministic systems and a special motion of nonlinear dynamic systems widely existing in nature [14]. Due to the defects of the classical chaotic systems, such as simple algebraic structure and few control parameters, the system output has a short period and the value distribution of the sequence is not uniform [15,16]. Many scholars are devoted to improving the complexity of classical chaotic systems. Zhou et al. [17] proposed a parametric switching chaotic system and applied it to image encryption schemes. Hua et al. [18,19] took the output of a low-dimensional system as the input of a nonlinear function to improve the dynamic characteristics of low-dimensional chaotic system. All these works were based on mathematical concepts without physical background. In 2008, Chua [20] replaced the original Chua diode with an active memristor in a chaotic circuit, and the results show that the circuit can also generate chaotic behavior. Later, scholars have done a lot of research on the memristor chaotic system, such as multi-stable states [21,22], co-existence attractor [23,24], and hidden attractor [25]. However, all these work are based on continuous-time memristor models (CMM). Compared with CMM, discrete-time memristor models (DMM) have not received much attention. In recent years, Peng et al. [26] proposed a discrete-time ideal charge-controlled memristor model. The output of the model was used as a control parameter of the two-dimensional Hénon mapping, and the output of the Hénon mapping was disturbed to enhance the dynamic characteristics of the Hénon mapping. Bao et al. [27] proposed a discrete time ideal charge-controlled memristor model, and linearly coupled the model with a logistic map to obtain a new two-dimensional memristor model. Deng et al. [28] also proposed a two-dimensional hyperchaotic system based on a discrete memristor and analyzed the nonparameteric bifurcation mechanism of the system. However, the above works just replace the output of the discrete memristor with the parameters of the chaotic system, or use the output of the discrete memristor as the input of the chaotic system. This method can improve the complexity of the chaotic system, but the effect is not obvious enough. In this paper, a generalized two-dimensional hyperchaotic system based on a discrete memristor is proposed. The discrete memristor is coupled with a classical logistic map, a sine map, and a tent map, respectively, to generate a two-dimensional memristor chaotic map. These three models are chaotic in a wide range of parameters, and even hyperchaotic in a suitable range of parameters. This method is simple and effective to enhance the dynamic characteristics of chaotic systems.

The rest of this paper is arranged as follows: In Section 2, the $TiO_{2}$ memristor model is discretized by using the backward-Euler method, and the discretized model also conforms to the essential characteristics of the generalized memristor. Then, the $TiO_{2}$ memristor model is linearly coupled with the one-dimensional chaotic system to obtain the two-dimensional discrete memristor map. In Section 3, three numerical examples are given and the three systems are evaluated by chaotic dynamics characteristic index and complexity analysis. In Section 4, three systems are used as chaotic sequence generators of chaotic keying communication scheme, and the advantages and disadvantages of chaotic sequence are evaluated by the bit error rate of communication scheme. Finally, some conclusions are draw in Section 5.

## 2. Two Dimensional Generalized Discrete Memristor Coupling Model

In 2014, Chua [29] proposed three mathematical expressions of time-varying memristor (i.e., CMM), namely three different memristor models. At the same time, according to the input signal of the memristor, each model is divided into charge-controlled memristor and flux-controlled memristor. The mathematics expression of these are shown in Table 1, where M(·) is the memristance function based on the charge-controlled memristor, while W(·) is the memristance function based on the magnetic flux-controlled memristor.

In the ideal model, M(·) is only a function of charge *q*, and the derivative of charge *q* is the current *i* through the device, while W(·) is only a function of magnetic flux ϕ. The derivative of magnetic flux ϕ is the voltage *v* at both ends of the device. In the general model, M(·) is only a function of the system state variable *x*, and the derivative of the system variable *x* is a function of the state variable *x* and the current *i* flowing through the device, denoted by f(x,i), while W(·) is also a function of the system state variable *x*, and the derivative of the system variable *x* is a function of the state variable *x* and the voltage *v* both ends of the device, denoted by g(x,v). In the generalized model, M(·) is the function of the system state variable *x* and the current *i* flowing through the device, and the derivative of the system variable *x* is a function of the state variable *x* and the current *i* flowing through the device, denoted by f(x,i), while W(·) is the function of the system state variable *x* and the voltage *v* at both ends of the device, and the derivative of the system variable *x* is a function of the state variable *x* and the voltage *v* both ends of the device, denoted by g(x,v). In this paper, we take the ideal charge-controlled memristor model as the research object, so we can get a charge-controlled memristor characterized by voltage v(t) and current i(t), and the mathematics expression is as follows
(1)$$\left\{\begin{array}{c} \;v(t)=M(q)i(t)\\ \;dq(t)/dt=i(t) \end{array}\right.$$
where v(t) and i(t) is the voltage and the input current of the memristor, and q(t) is the charge the memristor at time *t*. The relationship between charge q(t) and input current i(t) is written as
(2)$$q\left(t\right)=q\left(t_{0}\right)+\int_{t_{0}}^{t}i\left(\tau \right)d\tau ,$$
where $q(t_{0})$ is the initial charge of memristor. After discretization by the backward- Euler method, the mathematical expression of the discretized model is as follows:(3)$$\left\{\begin{array}{c} \;v_{n}=M\left(q_{n}\right)i_{n}\\ \;q_{n+1}=q_{0}+h\sum_{j=1}^{n}i_{j} \end{array}\right.$$
where *h* represents the step size required for each iteration. Generally, suppose h=1. In this paper, the memristance function $M(q_{n})$ is based on $TiO_{2}$ memristor model, and the mathematical expression of $M(q_{n})$ for the function can be obtained as
(4)$$M\left(q_{n}\right)=\left(R_{1}q_{n}+R_{2}{q_{n}}^{2}\right),$$
where $R_{1}=R_{OFF}$, $R_{2}=R_{OFF}\mu_{V}R_{ON}/D^{2}$. However, $R_{OFF}$ is the undoped partial voltage, $R_{ON}$ is the doped partial voltage, *D* is the length of the memristor element, and $\mu_{V}$ is the average ion mobility. In this paper, set $R_{1}=1,R2=0.001$. In order to illustrate the characteristics of the discrete-time $TiO_{2}$ memristor model, a discrete sine current signal $i_{n}=A\sin\left(\omega t\right)$ is added to the memristor. The amplitude curves of current $i_{n}$ and voltage $v_{n}$ of the memristor are shown in Figure 1. Given A,ω, and $q_{0}$ in details, the iterative sequences of current $i_{n}$ and voltage $v_{n}$ are shown in Figure 1a. By fixing $A,q_{0}$, the pinched hysteresis loop of the memeristor at different frequencies ω are simulated in Figure 1b. It can be seen that the area of the pinched hysteresis loop decreases with the increase of frequencies ω. By fixing $\omega ,q_{0}$, the pinched hysteresis loop of the memeristor at different amplitude *A* are plotted in Figure 1c. In addition, by fixing A,ω, the pinched hysteresis loop of the memeristor at initial charge *q* is depicted in Figure 1d. The above simulation results show that the discrete memristor model meets the characteristics of a generalized memristor.

### Memristor Coupled Low-Dimensional Discrete Chaos Model

Due to its simple algebraic structure, one-dimensional discrete chaotic systems such as logistic map, sine map, and tent map have defects such as short period phenomenon, uneven distribution of output sequence, and predictable motion trajectory. However, these defects will limit the application of one-dimensional chaotic systems in some security fields. In order to enhance the chaotic dynamics characteristics of one-dimensional chaotic maps, a generalized two-dimensional discrete memristor model (2D-DMM) is obtained by coupling one-dimensional chaotic systems with the discrete memristor model constructed above. The structural block diagram of 2D-DMM is shown in Figure 2, in which $F(\mu ,x_{n})$ represents an existing 1D discrete map, and μ is the control parameter of it. The parameter *k* is the coupling coefficient between the chaotic map and the discrete memristor model. Some 2D-DMM can be yielded by using such a system structure. The mathematical expression of 2D-DMM can be obtained
(5)$$\left\{\begin{array}{c} \;x_{n+1}=F\left(\mu ,x_{n}\right)+k\left(R_{1}q_{n}+R_{2}{q_{n}}^{2}\right)x_{n}\\ \;q_{n+1}=x_{n}+q_{n} \end{array}\right.$$

According to Equation (Equation 5), based on different one-dimensional discrete chaotic systems, corresponding two-dimensional discrete memristor models can be obtained by using this system framework. This model combines the outputs of two nonlinear systems linearly to improve the dynamics of one system. This method is simple and effective to enhance the dynamic characteristics of chaotic systems.

The stability of discrete chaotic system is described by its fixed point. A fixed point of a map is the value of iteration at the next moment, which is the same as the value at this moment. The fixed point of Equation (Equation 5) is denoted as $\left(\tilde{x},\tilde{q}\right)$, and one can obtain the following expression.
(6)$$\left\{\begin{array}{c} \;{\tilde{x}}_{n}=F\left(\mu ,{\tilde{x}}_{n}\right)+k\left(R_{1}{\tilde{q}}_{n}+R_{2}{{\tilde{q}}_{n}}^{2}\right){\tilde{x}}_{n}\\ \;{\tilde{q}}_{n}={\tilde{x}}_{n}+{\tilde{q}}_{n} \end{array}\right.$$
Obviously, when F(μ,x)=0, the system has infinite fixed points, which are expressed as
(7)$$S=\left(\tilde{x},\tilde{q}\right)=\left(0,\xi \right)$$
where ξ is an arbitrary real number. While F(μ,x)≠0, since Equation (Equation 6) has no real root, the system has no fixed point. The Jacobian matrix of Equation (Equation 5) is shown as follows
(8)$$J=\left[\begin{array}{cc} k\left(R_{1}q_{n}+R_{2}{q_{n}}^{2}\right)+\phi \left(\mu \right), & k\left(R_{1}q_{n}+R_{2}{q_{n}}^{2}\right)x_{n}\\ 1, & 1 \end{array}\right],$$
where $\phi \left(\mu \right)=dF\left(\mu ,x_{n}\right)/dx_{n}$. The eigenvalue polynomial at the fixed point *S* of Equation (Equation 5) can be obtained
(9)$$P\left(\lambda \right)=\left(\lambda -1\right)(\lambda -k\left(R_{1}q_{n}+R_{2}{q_{n}}^{2}\right)-\phi \left(\mu \right)).$$
Therefore, the two eigenvalues at fixed points *S* of Equation (Equation 5) are calculated as
(10)$$\lambda_{1}=1,\lambda_{2}=k\left(R_{1}q_{n}+R_{2}{q_{n}}^{2}\right)-\phi \left(\mu \right).$$
According to the stability criterion of fixed points, when the modules of the two eigenvalues are in the unit circle, the fixed point is stable, otherwise it is unstable. Since $\lambda_{1}$ is always on the unit circle, whether $\lambda_{2}$ is inside or outside the unit circle depends on ϕ(μ), the coupling coefficient *k*, and the initial state ξ of the memristor. Therefore, the fixed points *S* of the model may be stable or unstable. While F(μ,x)≠0, the Equation (Equation 6) has no fixed point. In this case, the trajectory of the map is periodic, chaotic, or even hyperchaotic.

## 3. Numerical Example of Two-Dimensional Discrete Memristor Model

In this Section, we construct the corresponding two-dimensional discrete memristor model based on the logistic map, sine map, and tent map, respectively. In addition, the performance of their chaotic sequences will be evaluated.

### 3.1. Numerical Examples

According to Equation (Equation 5), 2D-DMM based on the logistic map, sine map, and tent map can be obtained, respectively. The mathematical expression of the two-dimensional discrete memristor coupled-Logistic model (2D-DMLM) is as follows
(11)$$\left\{\begin{array}{c} \;x_{n+1}=\mu x_{n}\left(1-x_{n}\right)+k\left(R_{1}q_{n}+R_{2}{q_{n}}^{2}\right)x_{n}\\ \;q_{n+1}=x_{n}+q_{n} \end{array}\right.$$
where μ is the control parameter of the logistic map, F(μ,x)=0, and ϕ(μ)=μ. The mathematical expression of the two-dimensional discrete memristor coupled-Sine model (2D-DMSM) is described as
(12)$$\left\{\begin{array}{c} \;x_{n+1}=\mu \sin\left(2\pi x_{n}\right)+k\left(R_{1}q_{n}+R_{2}{q_{n}}^{2}\right)x_{n}\\ \;q_{n+1}=x_{n}+q_{n} \end{array}\right.$$
where μ is the control parameter of the sine map, F(μ,x)=0, and ϕ(μ)=2πμ. The mathematical expression of the two-dimensional discrete memristor coupled-Tine model (2D-DMTM) is obtained as
(13)$$\left\{\begin{array}{c} \;x_{n+1}=\left\{\begin{array}{c} \mu x_{n}+k\left(R_{1}q_{n}+R_{2}{q_{n}}^{2}\right)x_{n},x_{n}<0.5\\ \mu \left(1-x_{n}\right)+k\left(R_{1}q_{n}+R_{2}{q_{n}}^{2}\right)x_{n},x_{n}\geq 0.5 \end{array}\right.\\ \;q_{n+1}=x_{n}+q_{n} \end{array}\right.$$
where μ is the control parameter of the tent map, F(μ,x)=0, and ϕ(μ)=±μ. The mathematical expressions of the three models are given in this section, and the chaotic performance evaluation of the three models is given in the following section.

### 3.2. Bifurcation with Coupling Strength

The bifurcation diagram of a chaotic system is an important tool to analyze its characteristics. It can directly observe the period-doubling bifurcation and chaotic state of the system under different control parameters. The Lyapunov exponent (LE) is an important index to measure the separation rate of adjacent trajectories in phase space. In this paper, Wolf’s Jacobian algorithm is used to calculate LE. The LE can be calculated as follows
(14)$$LE=\lim_{n\to +\infty }\frac{1}{n}\sum_{i=0}^{n-1}\ln(\lambda_{i}),$$
where *i* represents the number of iterations of the system, while $\lambda_{i}$ is the eigenvalue of the Jacobian matrix of the system after *i* iterations. If a dynamic system has a positive Lyapunov exponent, the system is in chaos. The larger the value of LE is, the stronger the chaotic characteristics of the system will be. When the number of positive LE is greater than 1, the system is a hyperchaotic system, indicating that the system has stronger chaotic characteristics. Given control parameter μ and initial conditions $\left(x_{0},q_{0}\right)$, the bifurcation diagram and LE spectrum of the change of coupling coefficient *k* of the models are shown in Figure 3. As can be seen from Figure 3, under different coupling coefficients, the three models exhibit complex dynamic properties, such as periodic Windows, periodic or period-like states, chaos and hyperchaotic states. It is worth noting that 2D-DMLM and 2D-DMTM enter the chaos state from a period-doubling way, while 2D-DMSM enters chaos state from periodic state in period-like way. It can be seen from the LE spectrum that the system has a positive LE in most parameter intervals, and even two positive LEs in some intervals. Given the control parameter μ, initial conditions $\left(x_{0},q_{0}\right)$, and coupling coefficient *k*, the two Lyapunov exponents for the three models are given in Table 2. Note that the length of the sequence is set to $10^{5}$. It can be seen that the three models all have two positive LEs, indicating that the three models are all hyperchaotic.

### 3.3. Hyperchaotic Attractor

From Section 3.2, we know that the three models are in a hyperchaotic state with determined parameters. This section will study the hyperchaotic sequences generated by three models. Given control parameter μ, initial conditions $\left(x_{0},q_{0}\right)$, and coupling coefficient *k* in details, respectively, the trajectories of the three models in phase space are shown in Figure 4. It can be seen that all hyperchaotic attractors are distributed in a bounded region and have complex fractal structures. Secondly, hyperchaotic sequences generated by the three models are shown in Figure 5. The three sequences are all similar to noise signals, and the magnitude of sequence values is irregular and aperiodic. Next, we will evaluate the performance of hyperchaotic sequences through Sample entropy (SE) [30], Permutation entropy (PE) [31], Correlation dimension (CorDim) [32], and Kaplan–Yorke dimension (K-YDim) [33]. Note that the length of the sequence is set to $10^{5}$. The calculation results of the three hyperchaotic sequences are shown in Table 3. It can be seen that the hyperchaotic sequence generated by the system has excellent complexity and can be used in image encryption, chaotic secure communication, and other fields. Secondly, the discrete memristor model based on chaotic systems can enhance the complexity of a one-dimensional chaotic system.

## 4. Application of Hyperchaotic Sequence in Secure Communication

Due to the unpredictability and ergodicity of chaotic systems, chaotic systems are widely used to transmit data safely through various networks. When a chaotic system is used for data transmission, the distribution of its output has great influence on the performance of transmission error resistance. In this section, we take the three chaotic sequences as the chaotic sequence generators of the chaotic keying communication model and evaluate the sequences by the bit error rate of signal in the process of transmission with noise. In this paper, a chaotic keying communication model is used as the reference modulation-differential chaotic shift keying model (RM-DCSK). The RM-DCSK scheme consists of a transmitter and a receiver. The transmitter first uses a chaotic sequence to encode the information bits to generate the transmission signal, and then sends the transmission signal to the receiver. The receiver decodes the received signal to recover bits of information.

### 4.1. Transmitter Sructure

The transmission signal structure in RM-DCSK is depicted in Figure 6, where a frame signal is made up with two neighboring slots. The detailed structure of the sender of the RM-DCSK is plotted in Figure 7. Take the *k* frame signal for example, where the chaotic sequence $x_{k}$ is sent on the basis of information bit $b_{2k}\in \left\{-1,+1\right\}$ in the first slot. However, the signal of the second time slot is the sum of two parts, the first part is the modulation of information bit $b_{2k+1}\in \left\{-1,+1\right\}$ and the transmission signal of the first time slot; the second part is the chaotic sequence of this slot. We set the length of chaotic sequence to *M*, and the mathematical expression of the *k*-frame signal $s_{i}$ is as follows
(15)$$s_{i}=\left\{\begin{array}{c} b_{2k}x_{i},2kM<i\leq \left(2k+1\right)M\\ b_{2k+1}b_{2k}x_{i-M}+x_{i},\left(2k+1\right)M<i\leq 2\left(k+1\right)M \end{array}\right.$$
where $x_{i}$ satisfies the following conditions
(16)$$x_{(2k+1)M+m}=x_{2(k+1)M+m},0<m\leq M;k\in Z.$$

### 4.2. Receive Structure

When the receiver receives a transmission signal from the sender, the correlator can be used to recover the original bits of information. The RM-DCSK system mainly utilizes chaotic delay characteristics and noncoherent demodulation. The structure diagram of the receiving end is shown in Figure 8. Since signals may be interfered by noises when transmitted in different networks, the received signals are different from the original signals. In this paper, the Additive White Gaussian Noise (AWGN) channel model is used as the main communication transmission medium. The interference in the channel is mainly the noise signal ζ, so the received signal $r_{i}$ is as follows:(17)$$r_{i}=s_{i}+\zeta_{i}.$$
The output of the correlator $Z_{n}$ is the product of the received signal $r_{i}$ and its delayed version $r_{i-M}$. Therefore, the mathematical expression of correlator $Z_{n}$ is calculated as the sum
(18)$$Z_{n}=\sum_{i=nM+1}^{(n+1)M}r_{i}r_{i-M}.$$
Then, the mathematical expression of correlator $Z_{2k}$ for recovering bit $b_{2k}$ is rewritten as the sum
(19)$$\begin{array}{cc} Z_{2k} & =\sum_{i=2kM+1}^{(2k+1)M}\left(s_{i}+\zeta_{i}\right)\left(s_{i-M}+\zeta_{x_{i-M}}\right)\\ \; & =\sum_{i=2kM+1}^{(2k+1)M}\left(b_{2k-1}b_{2(k-1)}x_{i-2M}+x_{i-M}+\zeta_{i-M}\right)\\ \; & \times (b_{2b}x_{i}+\zeta_{i}). \end{array}$$
Equation (Equation 16) tells us that $x_{i-M}=x_{i}$, then Equation (Equation 19) can be simplified as
(20)$$\begin{array}{cc} Z_{2k} & =\sum_{i=2kM+1}^{(2k+1)M}\left(b_{2k-1}b_{2(k-1)}x_{i-2M}+x_{i-M}+\zeta_{i-M}\right)\\ \; & \times (b_{2b}x_{i}+\zeta_{i})\\ \; & =b_{2k}\sum_{i=2kM+1}^{(2k+1)M}{x_{i}}^{2}+\sum_{i=2kM+1}^{(2k+1)M}x_{i}A_{i} \end{array}$$
where $A_{i}$ is calculated as follows:(21)$$\begin{array}{cc} Z_{2k} & =b_{2k}b_{2k-1}b_{2(k-1)}x_{i}x_{i-2M}+b_{2k}x_{i}\zeta_{i-M}\\ \; & +b_{2b-1}b_{2(b-1)}x_{i-2M}\zeta_{i}+x_{i}\zeta_{i}+\zeta_{i}\zeta_{i-M}. \end{array}$$

Since the noise signal needs much less energy than the useful signal, the square term containing chaotic signal is the signal of subject discrimination. Then the positive and negative of the correlator $Z_{2k}$ is mainly determined by the bit information $b_{2k}$. In addition, the correlator $Z_{2k+1}$ can similarly recover information bit $Z_{2k+1}$. Finally, information bit $Z_{n}$ will be demodulated according to the decision threshold.
(22)$$b_{n}=\left\{\begin{array}{cc} 1, & b_{n}>0\\ -1, & b_{n}\leq 0 \end{array}\right..$$
where $x_{i}$ satisfies the following conditions
(23)$$x_{(2k+1)M+m}=x_{2(k+1)M+m},0<m\leq M;k\in Z.$$

### 4.3. Simulation Results

In order to display the complex dynamic characteristics of the generated hyperchaotic sequence, we use three groups of chaotic models as the RM-DCSK scheme chaotic sequence generators. The chaos model of the first group is the classical logistic map, sine map, and tent map. The second group was the memristor model proposed in reference [27]. We generated two-dimensional memristor models based on the classical logistic map, sine map, and tent map, denoted as 2D-MLM, 2D-MSM, and 2D-MTM, respectively. The last group is three two-dimensional memristor models constructed in this paper. We use the above three groups of chaotic sequence generators to simulate the RM-DCSK scheme. The bit error rates (BERs) are calculated under different signal-noise-rate (SNR) and spread spectrum factor M. Note that each experiment transmits data in a randomly generated binary sequence of $10^{5}$ bits. The initial conditions and control parameters of these models are listed in Table 4, which can make these models in a state of chaos.

Firstly, the RM-DCSK scheme with different SNR are considered. By setting SNR ∈{0,1,2,⋯,30} and fixing M =300, we simulated the RM-DCSK scheme using three groups of chaotic sequence, respectively. The BERs of the demodulated and modulated signal under different SNR is shown in Figure 9a. It can be seen that when the spectrum factor M is determined, when SNR is small, the models based on three groups of chaotic sequence generators can obtain almost the same BERs. However, the model based on the third group of the chaotic sequence can obtain smaller BERs than the first two groups of chaotic sequence with the increase of SNR. Furthermore, By setting M ∈{260,260,270,⋯,350} and fixing SNR =18, we also simulated the RM-DCSK scheme using three groups of chaotic sequence, respectively. The BERs of the demodulated and modulated signal under different spectrum factor M is depicted in Figure 9b. As shown, the model based on the third group of chaotic sequence can obtain smaller BERs than the first two groups of chaotic sequence under different M. Chaotic system is used in secure communication. The distribution of chaotic output sequence is a key factor affecting transmission efficiency. Therefore, compared with the first two groups of chaotic sequence generators, the third group of chaotic sequence generators is more suitable for chaotic secure communication.

## 5. Conclusions

As the fourth basic circuit component, the memristor is a nonlinear element. If the memristor is introduced into the chaotic circuit, the complexity of the circuit can be strengthened. In this paper, we discretized the ideal flow-controlled memristor model using the Eluer method, obtained the discrete memristor model, and coupled the discrete memristor model with the one-dimensional discrete chaotic system to obtain the two-dimensional discrete memristor coupling model. In chapter 3, three real numerical examples are given. The Lyapunov index, bifurcation diagram and chaotic hyacinth are used to show that the dynamic behavior of the system has been significantly improved. Secondly, the system has high complexity through sample entropy, permutation entropy, correlation dimension, and the Kaplan–Yorke dimension. Finally, the model proposed in this paper is applied to secure communication. Compared with the one-dimensional chaotic system and the existing two-dimensional memristor model, the transmission efficiency of the model proposed in this paper is significantly improved. Therefore, by coupling the model with the discrete memristor, the dynamic characteristics of the low-dimensional chaotic system can be enhanced. 

## Figures and Tables

**Figure 1 entropy-24-00864-f001:**
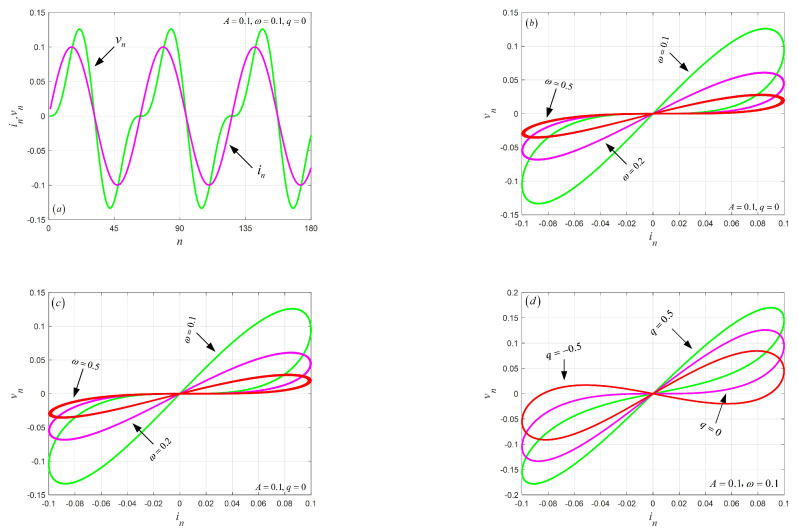
The $i_{n}-v_{n}$ amplitude curves of the memeristor: (**a**) $i_{n}$ and $v_{n}$ iterative sequences; (**b**) The pinched hysteresis loop of the memeristor at different frequencies ω; (**c**) The pinched hysteresis loop of the memeristor at different amplitudes *A*; (**d**) The pinched hysteresis loop of the memeristor at different initial charges *q*.

**Figure 2 entropy-24-00864-f002:**
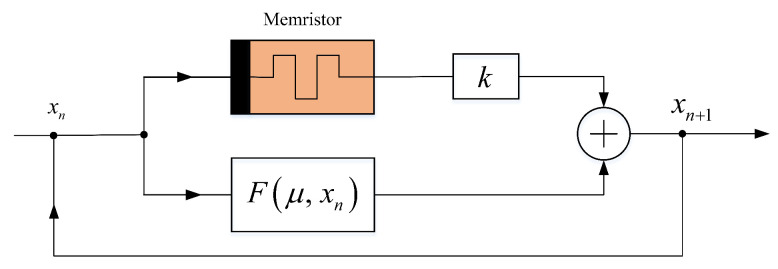
Structure block diagram of the two-dimensional discrete memristor model.

**Figure 3 entropy-24-00864-f003:**
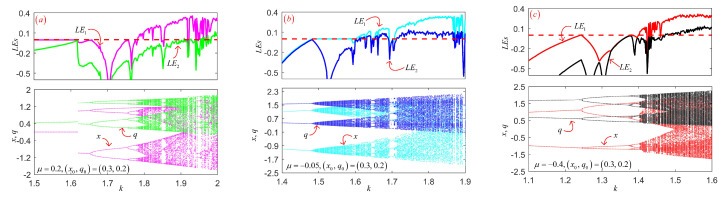
The bifurcation and LEs of three models: (**a**) 2D-DMLM; (**b**) 2D-DMSM; (**c**) 2D-DMTM.

**Figure 4 entropy-24-00864-f004:**
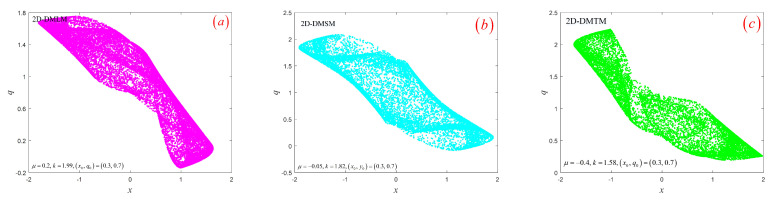
The phase space trajectory of three models: (**a**) 2D-DMLM; (**b**) 2D-DMSM; (**c**) 2D-DMTM.

**Figure 5 entropy-24-00864-f005:**
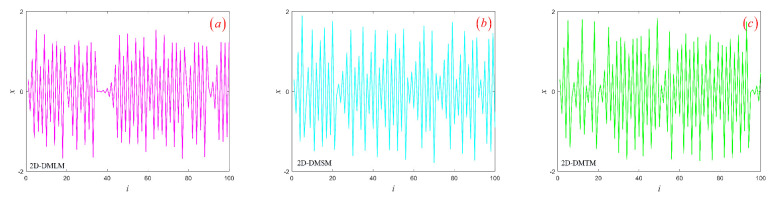
The hyperchaotic sequence produced by three models: (**a**) 2D-DMLM; (**b**) 2D-DMSM; (**c**) 2D-DMTM.

**Figure 6 entropy-24-00864-f006:**
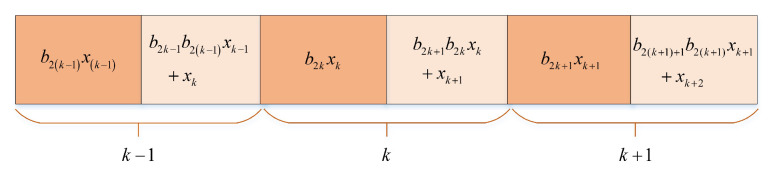
Structure of the transmission signal in RM-DCSK.

**Figure 7 entropy-24-00864-f007:**
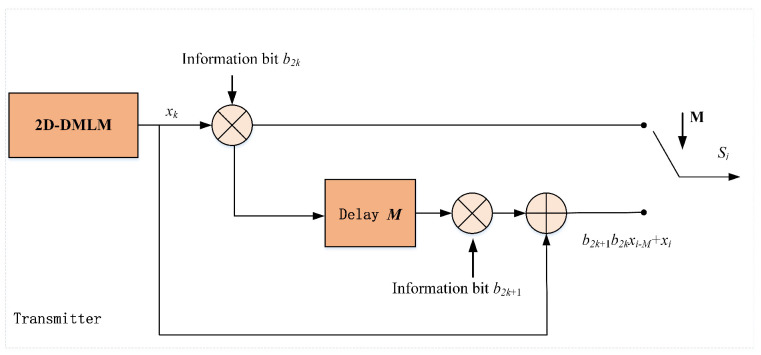
Structure of the transmission signal in RM-DCSK.

**Figure 8 entropy-24-00864-f008:**
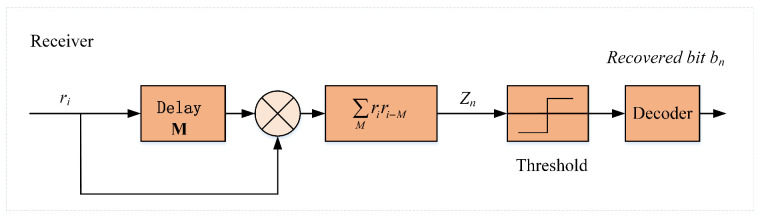
Structure of the transmission signal in RM-DCSK.

**Figure 9 entropy-24-00864-f009:**
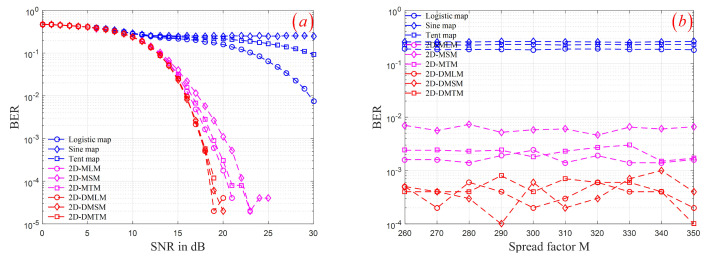
BERs of the RM-DCSK using three groups of chaotic sequence: (**a**) different SNR; (**b**) different M.

**Table 1 entropy-24-00864-t001:** Mathematical models of three continuous-time memristors.

CMs	Charge-Controlled	Flux-Controlled
ideal	$\left\{\begin{array}{c} v(t)=M(q)i(t)\\ dq/dt=i(t) \end{array}\right.$	$\left\{\begin{array}{c} i(t)=W(\phi )v(t)\\ d\phi /dt=v(t) \end{array}\right.$
general	$\left\{\begin{array}{c} v(t)=M(x)i(t)\\ dx/dt=f(x,i) \end{array}\right.$	$\left\{\begin{array}{c} i(t)=W(x)v(t)\\ dx/dt=v(x,v) \end{array}\right.$
generalized	$\left\{\begin{array}{c} v(t)=M(x,i)i(t)\\ dx/dt=i(x,i) \end{array}\right.$	$\left\{\begin{array}{c} i(t)=W(x,v)v(t)\\ dx/dt=v(x,v) \end{array}\right.$

**Table 2 entropy-24-00864-t002:** The two Lyapunov exponents of three models.

Iterms	μ	*k*	$\left(x_{0},q_{0}\right)$	${\mathrm{LE}}_{1},{\mathrm{LE}}_{2}$
2D-DMLM	0.200	1.99	(0.3, 0.7)	0.3028, 0.0731
2D-DMSM	−0.05	1.82	(0.2, 0.4)	0.3200, 0.0757
2D-DMTM	−0.40	1.58	(0.1, 0.6)	0.2797, 0.1062

**Table 3 entropy-24-00864-t003:** Performance of hyperchaotic sequences generated by three models.

Iterms	SE	PE	CorDim	K-YDim
2D-DMLM	0.7268	0.8311	1.6163	2
2D-DMSM	0.7786	0.8010	1.6315	2
2D-DMTM	0.7076	0.8216	1.6295	2

**Table 4 entropy-24-00864-t004:** The initial conditions and control parameters of these models.

Items	Initial Conditions	Control Parameters
Logistic	$x_{0}=0.4$	μ=3.87
Sine	$x_{0}=0.2$	μ=0.95
Tent	$x_{0}=0.3$	μ=1.47
2D-MLM	$x_{0}=0.5,\;q_{0}=0.5$	μ=0.1,k=1.88
2D-MSM	$x_{0}=0.4,\;q_{0}=0.7$	μ=−0.1,k=1.78
2D-MTM	$x_{0}=0.3,\;q_{0}=0.6$	μ=−0.3,k=1.68
2D-DMLM	$x_{0}=0.2,\;q_{0}=0.3$	μ=0.2,k=1.99
2D-DMLM	$x_{0}=0.4,\;q_{0}=0.1$	μ=−0.05,k=1.82
2D-DMTM	$x_{0}=0.2,\;q_{0}=0.6$	μ=−0.4,k=1.58

## Data Availability

Data sharing not applicable to this article, as no datasets were generated or analyzed during the current study.

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
