# Peer review of "Discrete-Time Memristor Model for Enhancing Chaotic Complexity and Application in Secure Communication"

_entropy, 2022, doi:10.3390/e24070864_

Round 1

Reviewer 1 Report

The article Discrete-time Memristor Model for Enhancing Chaotic Complexity and Application in Secure Communication is an interesting approach of combining the memristor complex dynamics with another discrete chaotic systems. It has been proved that the above combination significantly increases the final complexity of the system as also hyperchaotic phenomenon has been noticed. Authors pointed out the interesting application of the combined system to the image encryption and the chaotic communication. The second applications is also elaborated in the Chapter 4 of the article. This might attract the article readers to perform further analysis of this application.

 The weaker point is that the used TiO­2 model of the memristor is quite out dated and it is shown already that this model cannot be considered to study the real memristors behaviour. It should be mentioned about this acpect. On the other hand in the reviewer opinion the presented results are going to be similar and the final conclusions would be indistinguishable.

I would only remark some minor aspect to be improved in the final manuscript.

1.       TiO2 should have the 2 in the subscripted i.e. TiO2 (for example p.1 l. 3, 8, 10, 13, 20…..)

2.       Page 1 line 4 fingerpriter  -> fingerprints

3.       Please rewrite the sentence (p.1 line. 22): In 2013, …… signals, the device is a tight hysteresis loop… I guess authors meant that, on the v-i plane one can observe the hysteresis loop…

4.       Further, the mentioned critical frequency. I guess this memristor phenomenon that the lobe area decreases while frequency of the signal increases doesn’t not have any particular critical frequency. Please answer or amend the manuscript.

5.       P.1 line 29 – I am not sure about the statement of the randomness of the deterministic chaos. Chaos is not random as it is deterministic…. In the reference [14] it is also well explained.

6.       I have an issue with the memristor that has been studied. In p. 1 line 29 it should be mentioned that Itoh and Chua indeed replaced the Chua diode with the active memristor (please add word active memristor) forming the interesting simple circuit with very complex behaviour. Then on page 3 authors use the simple Strukov TiO­2 memristor model [3] with negative resistance (active element). It should be explained if it is necessary to apply the active memristor? This element doesn’t exist and the application of such study become weaker.

In the eqn. 4 authors set R­1=-2 and R2=1, form the Strukov model we can assume Roff=-1 Om and the Ron=-5mOm (setting standard D=10nm and \mu_v=10^{-14}).  And again this negative values creates some confusions.

Also fig. 1 proves that the studied element is active. Please elaborate and explain this aspect in the final manuscript.

7.       Equation 2 under the integral i(t)dt should be i(tau)d\tau – for the mathematical formality

8.       The step size h for the Euler method is constant for all the integration step. But the definition (right after the eqn. 3) …, h represents the step size required for each iteration – might suggest that is variable and so should have also index i. Please clarify this confusion.

9.       There is a mistake in the definition of \mu_v and D of the after the eqn  4. I is the other way around (D is the length, and \mu_v is the ions average mobility).

10.   In general the graphs (fig. 1, 2, 3 etc.)descriptions are very small, please increase significantly  the font size for better readability.

11.   Please explain why the graphs presented on fig. 1 are in iterative sequence but not in nominated units: s – time signals, V – voltages and A – amperes.  Explain why graphs on 1.a doesn’t correspond with for example graphs 1.c – the voltage values are between -0.5 and 0.5 and on the graphs c) of some thousands.

12.   On page 4 line 88 please correct and rewrite the sentence … which means…

13.   Page 6 line 104. Also correct the sentences For a chaotic system.(??) At least…..

14.   In Conclusion p. 10 line 180 – the please replace function with element.

If the above suggestions especially mentioned in point 6 are considered I recommend the article to be published in the Entropy Journal.

Author Response

Thanks for the expert opinion.

Reviewer 2 Report

This paper introduces the TiO2 discrete memristor model in three classical one-dimensional chaotic maps, and simulations demonstrate enhanced chaos dynamics of the memristor-based chaotic maps. The presentation of this paper is clear and well structured. I suggest it can be accepted. However, before the paper is further processed, the authors need to consider the following issues:

1) The authors should clarify the motivation of this paper. In the introduction, the authors claim that "However, the above work is only based on a single low-dimensional discrete chaotic system", I don't think this is a good academic motivation. Futhermore, I recommend that the authors should give a broader view on the topic of discrete memristor and the current state-of-the-art.

2) I suggest that the authors put the fixed point analysis in the Section 3 and make a specific analysis of the three systems.

3) There are some typos in the titles of Fig. 3, Fig. 4 and Fig. 5.

4) English should be more polished, like "The larger the value of LE is, the stronger the chaotic characteristics of the system will be", the tenses in the sentence are inconsistent. The authors should double check the paper to avoid similar problems.

Author Response

Thanks for the expert opinion.
